# The interplay between Wnt and mTOR signaling modulates ciliogenesis in human retinal epithelial cells

Cheng Yuan[1,2], Annett Neuner[3], Johanna Streubel[1,2], Ayushi Bhanushali[4], Matias Simons[5], Sergio P. Acebrón[4,6,7], Gislene Pereira [1,2,3]*

1 Centre for Organismal Studies (COS), Cytoskeleton, Cell Division and Signal transduction Unit, University of Heidelberg, Heidelberg, Germany, 2 German Cancer Research Centre (DKFZ), Molecular Biology of Centrosome and Cilia Unit, DKFZ-ZMBH Alliance, Heidelberg, Germany, 3 Centre for Molecular Biology (ZMBH), University of Heidelberg, Heidelberg, Germany, 4 Centre for Organismal Studies (COS), Cell Signalling Unit, University of Heidelberg, Heidelberg, Germany, 5 Institute of Human Genetics, University Hospital Heidelberg, Heidelberg, Germany, 6 IKERBASQUE, Basque Foundation of Science, Bilbao, Spain, 7 University of the Basque Country (UPV/EHU), Leioa, Spain

* gislene.pereira@cos.uni-heidelberg.de, g.pereira@dkfz.de

## Abstract

The primary cilium is a microtubule-based organelle essential for various cellular functions, particularly signal transduction. While the role of cilia in regulating signaling pathways has been extensively studied, the impact of signaling pathways on cilia formation remains less well understood. Wnt signals are critical modulators of cell fate. In this study, we investigate how modulating Wnt signaling affects cilia formation in human retinal pigment epithelial (hTERT-RPE1) cells. Our findings show that enhancement of Wnt/LRP6 signaling before serum starvation delays ciliogenesis. Cells with high baseline Wnt activity exhibited distal appendage dysregulation, failure to remove CP110-CEP97 from mother centrioles, and reduced Rab8-vesicle docking, which are critical events for cilia membrane establishment and axoneme extension. Additionally, these cells displayed reduced autophagic flux, increased mTOR kinase activity, and elevated OFD1 levels at centriolar satellites. Importantly, mTOR inhibition rescued ciliogenesis in cells with elevated Wnt activity, underscoring the interplay between these signaling pathways. Our data also indicate that insufficient Wnt signaling activation disrupts ciliogenesis, emphasizing the need for precisely regulated Wnt levels.

## Introduction

Primary cilia are microtubule-based organelles that in vertebrate cells serve as key signaling hubs for signal transduction pathways, including Hedgehog, Hippo, TGF-β, Notch, and Wingless-related integration site (Wnt), among others [1]. Dysfunctional primary cilia can lead to a group of pleiotropic developmental and degenerative

**Data availability statement:** All relevant data are within the paper and its Supporting information files.

**Funding:** This work was funded by the German Research Foundation (Deutsche Forschungsgemeinschaft, DFG) SFB1324/B09 collaborative grant (granted to G.P. and M.S.). The funders had no role in study design, data collection and analysis, decision to publish, or preparation of the manuscript.

**Competing interests:** The authors have declared that no competing interests exist.

**Abbreviations:** APC, adenomatous polyposis coli; Co-CM, control-conditioned media; DAPI, 4′,6-diamidino-2-phenylindole; DKK1, Dickkopf-related protein 1; DVL, Disheveled; FBS, fetal bovine serum; FMCD, focal malformation of cortical development; FZD, Frizzled; IFT, intraflagellar transport; IMCD3, inner medullary collecting duct; LC3, light chain 3; NGS, next-generation surrogate; PBS, phosphate-buffered saline; S6K, S6 kinase; siRNAs, small interfering RNAs; U-ExM, ultrastructure expansion microscopy; Wnt, Wingless-related integration site.

diseases collectively known as ciliopathies. These disorders exhibit a broad range of clinical manifestations, such as skeletal and brain malformations, obesity, retinal degeneration, and polycystic kidneys [2].

Cilia biogenesis begins at the mother centriole of the centrosome, which converts into the basal body of the cilium during the G1/G0 phase of the cell cycle [3]. Cilia initiation involves the coordination of cytoskeleton rearrangements, membrane trafficking to the centrosome, and degradation of cilia-inhibitory components to facilitate ciliary membrane formation and axonemal microtubule extension starting from the basal body [4]. Key to this process are distal appendage components, which constitute a group of proteins that interact at the distal end of the mother centriole. They are essential for the docking of ciliary vesicles at the mother centriole by activating Rab-family GTPases, notably the Rab11-Rab8 cascade [4,5]. Additionally, distal appendages play a key role in the removal of the CP110-CEP97 inhibitory complex, which caps the end of the mother centriole, thereby preventing vesicle fusion and inhibiting ciliary growth [6,7]. Furthermore, distal appendages also promote axonemal microtubule extension by recruiting intraflagellar transport (IFT) protein complexes necessary for cilia growth [8–12].

Centriolar satellites are additional key regulators of ciliogenesis [13,14]. These membrane-less granules control the transport of centrosomal and cilia-related proteins, as well as regulatory components like E3-ubiquitin ligases, to the pericentriolar area via microtubules [13–15]. Many satellite components, such as the scaffolding protein PCM1 or WDR8, act as positive regulators of ciliogenesis and are required for the removal of the CP110-CEP97 complex and cilia vesicle establishment at the mother centriole [16–18]. However, the ciliopathy-related protein OFD1 localizes at centrioles but also at centriolar satellites, were it inhibits ciliogenesis by preventing the centrosomal recruitment of BBS4, an IFT protein essential for cilia biogenesis [19]. Notably, OFD1 is removed from centriolar satellites by autophagy during early steps of cilia formation [19].

Wnt is a highly conserved signaling pathway that governs various cellular processes, such as embryonic development, tissue regeneration, and cell fate determination [20,21]. The Wnt/β-catenin pathway, often termed as canonical, is activated by Frizzled (FZD) receptors and low-density-lipoprotein receptor related 5 or 6 (LRP5/6) co-receptors. Binding of Wnt ligands to FZD-LRP5/6, results in receptor clustering on Disheveled (DVL) platforms termed LRP6 signalosomes, and subsequent phosphorylation of LRP6 by GSK3β and CK1γ [22–25]. LRP6 signalosomes recruit the β-catenin destruction complex, which contains the scaffold proteins AXIN1 and adenomatous polyposis coli (APC), the kinases CK1α and GSK3β, and the E3-ubiquitin ligase β-TrCP [26]. LRP6 signalosomes can mature into multivesicular bodies, sequestering the Wnt receptors together with GSK3β for long-term signaling [27]. The best-established output of Wnt/LRP6 signaling is the stabilization of cytoplasmic β-catenin, which translocates to the nucleus and interacts with members of TCF family of transcription factors to activate the transcription of target genes [28,29].

In addition to the β-catenin programme, Wnt/LRP6 signaling modulates or stabilizes many other GSK3 target proteins [30]. For example, GSK3 phosphorylation of

target proteins is able to generate phospho-degrons that can be recognized by specific E3-ubiquitin ligases, leading to protein degradation [30–32]. Wnt-dependent stabilization of proteins (Wnt/STOP) reduces protein degradation by inhibiting GSK3, thereby inducing cell growth, notably during mitosis [31,33]. Wnt/LRP6 signaling can also promote cell growth via mTOR (Wnt/mTOR) by preventing the GSK3-dependent phosphorylation and activation of the mTORC1 inhibitor TSC2 [34].

Canonical Wnt signaling have been implicated in ciliogenesis, although results have been inconsistent. For instance, Wnt activation enhances ciliogenesis in some contexts, it inhibits or has no influence in others [35–39]. Here, we show that enhancing Wnt/LRP6 signaling in human retinal epithelial (RPE1) cells prior to serum starvation impairs the removal of cilia inhibitory components, including CP110-CEP97 complex and OFD1 at centriolar satellites. Both ectopic removal of OFD1 and chemical inhibition of mTOR rescued ciliogenesis in cells with higher baseline Wnt activity, highlighting the contribution of Wnt/mTOR to cilia formation. We propose that while Wnt is necessary for cilia formation, any imbalance leading to lower or higher baseline Wnt activity compromises ciliogenesis.

## Results

### Disruption of baseline Wnt activity impairs cilia formation in RPE1 cells

Although non-transformed RPE1 cells exhibit non-induced, basal Wnt/β-catenin activity [36,38,40], which has been linked to ciliogenesis, the consequences of blocking or activating Wnt signaling for cilia formation in these cells remain unclear [36–38]. To address the importance of this basal Wnt signaling for ciliogenesis, we depleted endogenous LRP5/6 and β-catenin using siRNA treatment prior to cilia induction by serum deprivation. Compared to mock depletion (siLUC), depletion of LRP5/6 (S1A Fig) or β-catenin (S1B Fig) significantly decreased ciliogenesis by 50%–30% (Fig 1A and 1B), indicating that basal Wnt/β-catenin activity in RPE1 cells is required for cilia formation, as previously suggested [36]. Interestingly, cilia formed in LRP5/6- or β-catenin-depleted cells elongated to a length similar to or slightly longer than those in control-treated cells, respectively (S1C and S1D Fig), suggesting that Wnt/β-catenin signaling may differentially control cilia initiation and elongation.

Next, we sought to explore the impact of raising baseline Wnt activity for cilia formation. For this, we constructed RPE1 cells stably carrying the 7xTGC Wnt reporter construct, which carries GFP under control of the Wnt Tcf responsive promoter in addition to mCherry under the constitute SV40 promoter [41]. In the absence of Wnt signaling, only mCherry expression was observed. The addition of Wnt3a-conditioned media (Wnt3a-CM), but not control-conditioned media (Co-CM), resulted in an increased number of cells exhibiting high GFP-Wnt signal reporter activity within 6 hours of Wnt3a-CM treatment (S2A Fig). Based on these data, RPE1 cells were treated with Co-CM or Wnt3a-CM six hours prior to serum starvation to increase Wnt baseline activity. The same treatment was repeated at the time of serum starvation to maintain Wnt activity throughout ciliogenesis, as depicted in Fig 1C. We confirmed that Wnt signaling remained active after 16 h (Fig 1D) and even 48 h (S2B Fig) of serum starvation by determining the levels of β-catenin, which increased as a consequence of its stabilization. In comparison to Co-CM, Wnt3a-CM treatment significantly reduced ciliation in RPE1 cells by 60% (Fig 1E). This inhibitory effect was still observed after 30 h in serum-free medium, yet not as pronounced as at earlier time points (S2C Fig). At later time points (54 h), no significant difference was observed, suggesting a delay rather than a complete block of ciliogenesis as a consequence of increased Wnt baseline activity.

Importantly, DNA-content analysis by FACS indicated that both Co-CM and Wnt3a-CM-treated cells were similarly arrested in the G1/G0 phase of the cell cycle (S2D Fig). In addition, the levels of phosphorylated Rb protein (S807/811), a marker for proliferating cells [42], was equally reduced in both Co-CM and Wnt3a-CM samples (Fig 1F), indicating that the negative effect of Wnt activation on ciliogenesis was not due to the inability of the cells to arrest at the G1/G0 phase of the cell cycle upon serum starvation. We also conducted a similar analysis using additional cell lines. We observed that Wnt overactivation also impaired cilia formation in murine fibroblasts (NIH3T3, S2E Fig) and murine inner medullary collecting

PLOS Biology

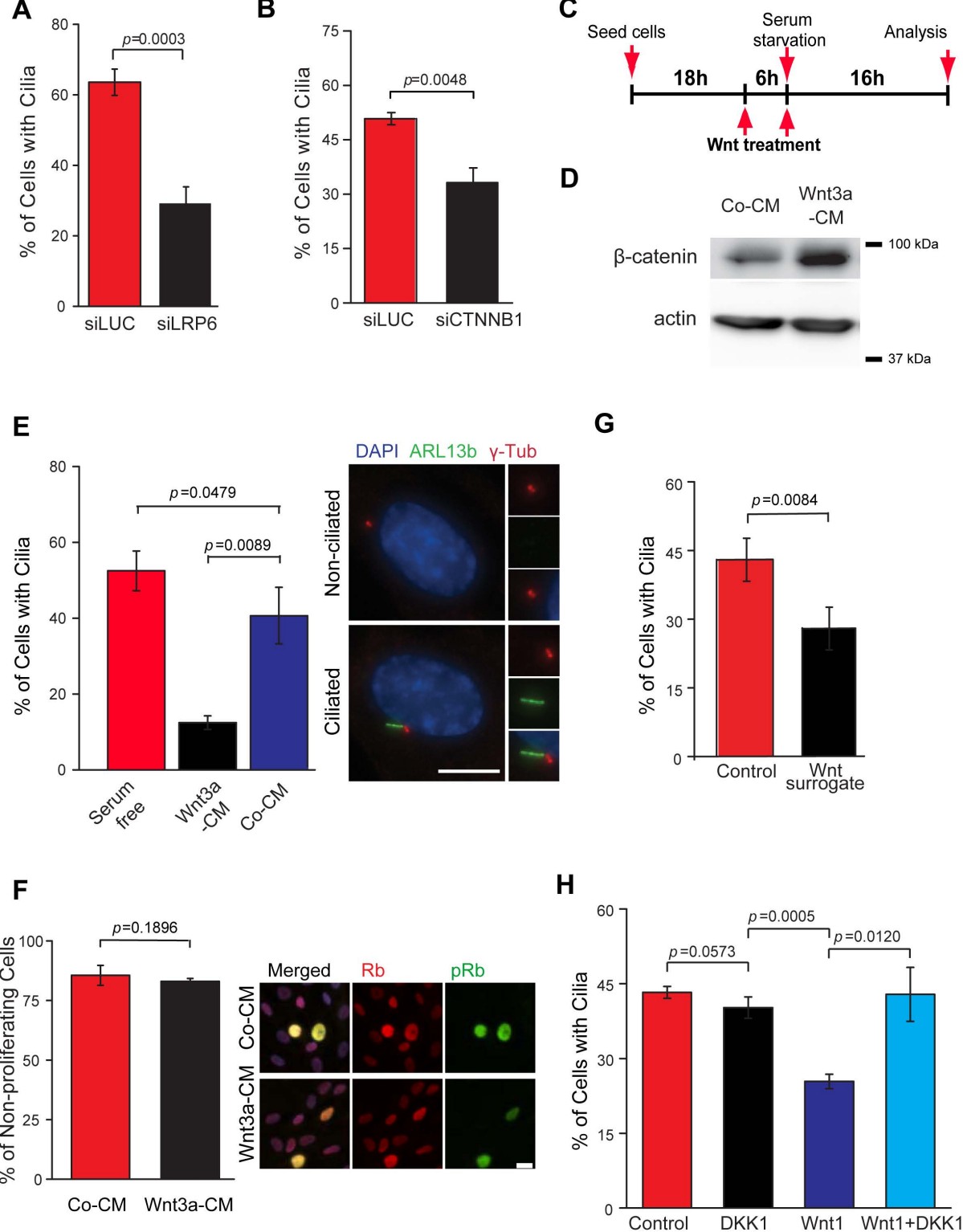

**Fig 1. Changes in baseline Wnt activity impair cilia formation in RPE1 cells. (A, B)** Quantification of ciliation of RPE1 cells treated with control (siLUC) and LRP6 (A) or β-catenin (B, CTNNB1) siRNA and serum starved for 24 h prior to fixation and staining with γ-tubulin (basal body marker) and

ARL13b (cilia membrane marker). The bar graph indicates mean ± S.D. from three independent experiments. (A) siLUC, $n = 561$; siLRP6, $n = 339$; (B) siLUC, $n = 312$; siCTNNB1, $n = 647$. **(C)** Experimental scheme for cilia induction and treatment of cells with Wnt agonist. **(D)** Western blot analysis of saponin lysed RPE1 cells treated with Co-CM or Wnt3a-CM (as depicted in C) and serum-starved for 16 h. Actin served as a loading control. **(E)** RPE1 cells were treated as described in (C) and subjected to immunofluorescence analysis 16 h after serum starvation. The bar graph indicates the percentages (mean ± S.D.) of ciliated cells treated with Co-CM and Wnt3a-CM from three independent experiments. Serum starved cells without treatment (serum free) were used as a control. The images show representative ciliated and non-ciliated cells with enlarged views of the centrosomal area. Basal bodies (γ-tubulin, red), ciliary membrane (ARL13b, green) and DNA (DAPI staining, blue) are shown as single channels or merged. Scale bar: 5 μm. **(F)** RPE1 cells were treated as described in (C) and stained with Rb and phospho-Rb (pRb; S807/S811) antibodies 16 h after serum-starvation. Representative images and quantifications of the percentage of non-proliferating cells (pRb negative) are shown. Co-CM, $n = 638$; Wnt3a-CM, $n = 876$. The bar graph indicates the mean ± S.D. from three independent experiments. Scale bar: 10 μm. **(G)** Quantification of ciliation in RPE1 cells treated with buffer control ($n = 561$) or purified surrogate Wnt agonist ($n = 639$) after 16 h of serum starvation. The bar graph indicates the mean ± S.D. from three independent experiments. **(H)** Quantification of ciliation after 16 h of serum starvation in RPE1 7xTGC cells expressing *DKK1* alone or in combination with *WNT1*. Empty plasmids were used as a control. The bar graph indicates the mean ± S.D. from three independent experiments. Control, $n = 559$; DKK1, $n = 597$; Wnt1, $n = 553$; Wnt1 + DKK1, $n = 612$. *P* values are based on Student *t* test. The data underlying the graphs and blots in this figure can be found in the S1 Data and S1 Raw Images files.

duct (IMCD3) cells (S2F Fig). Together, these data show that enhancing activation of Wnt signaling delays ciliogenesis in human and murine cells.

Wnt ligands, such as Wnt3a and Wnt1, activate the Wnt pathway through the dimerization of the Wnt receptor FZD and its co-receptors LRP5/6 [20]. This activation step is counteracted by the Dickkopf-related protein 1 (DKK1), which inhibits the association of Wnt ligand to LRP receptors [43]. To address whether the effect of Wnt on ciliogenesis requires FZD-LRP5/6, we first made use of a recombinant surrogate Wnt agonist, which induces heterodimerisation of FZD and LRP5 or LRP6 and hence Wnt activation, which can be suppressed by addition of DKK1 (S2G Fig) [44]. Adding purified recombinant Wnt surrogate to RPE1 cells was able to activate the Wnt pathway, as determined by appearance of GFP-positive cells using the reporter assay (S2H Fig). Similar to Wnt3a-CM, recombinant Wnt surrogate significantly diminished the percentage of ciliated RPE1 cells after 16 h of serum starvation (Fig 1G). However, neither Wnt3a-CM nor Wnt surrogate treatment affected the ciliary length of cells that were still able to form a cilium (S3A and S3B Fig). In a parallel assay, expression of *WNT1*, which activates Wnt/LRP signaling in RPE1 cells, also decreased the percentage of ciliated cells and this effect was counteracted by co-transfection of *DKK1* (Fig 1H). These data collectively indicate that high basal levels of Wnt/LRP signaling negatively affect ciliogenesis.

### Wnt overactivation delays cilia formation independently of TCF7

In the canonical Wnt signaling pathway, β-catenin translocates to the nucleus, where it interacts with TCF/LEF transcription factors to regulate the expression of Wnt target genes [28,29]. Notably, TCF7 has been shown to play a relevant role in RPE cells [45]. To test whether Wnt pathway overactivation influences ciliogenesis via TCF/LEF-dependent transcriptional regulation, we investigated whether TCF7 is required for ciliogenesis. First, we verified that in the vast majority of RPE1 cells (>99%), TCF7 accumulated in the nucleus upon Wnt3a addition (Fig 2A), confirming TCF7 responsiveness to Wnt. We next depleted TCF7 using small interfering RNAs (siRNAs) before exposing the cells to Co-CM or Wnt3a-CM (Fig 2B). In the presence or absence of TCF7, Wnt3a-CM increased the amount of slow-migrating, hyperphosphorylated forms of DVL2, which serves as a marker for Wnt activation (Figs 2B, lanes 2 and 4 and S3C) [46]. Wnt3a-CM similarly impaired cilia formation irrespective of the presence of TCF7 (Fig 2B and 2C), suggesting that TCF7 is not required for the negative effect of Wnt overactivation on ciliogenesis. However, cilia formed in the absence of TCF7 were longer compared to control-depleted samples, regardless of Wnt hyperactivation (Fig 2D), indicating that TCF7 might control ciliary length.

### Wnt signaling activation decreases the recruitment of RAB8a to basal bodies

GSK3 was shown to promote ciliary membrane extension in a RAB8a-dependent manner [35]. Indeed, the inhibition of GSK3 in RPE1 cells with either BIO or CHIR99021, two GSK3-specific inhibitors, significantly suppressed cilia formation

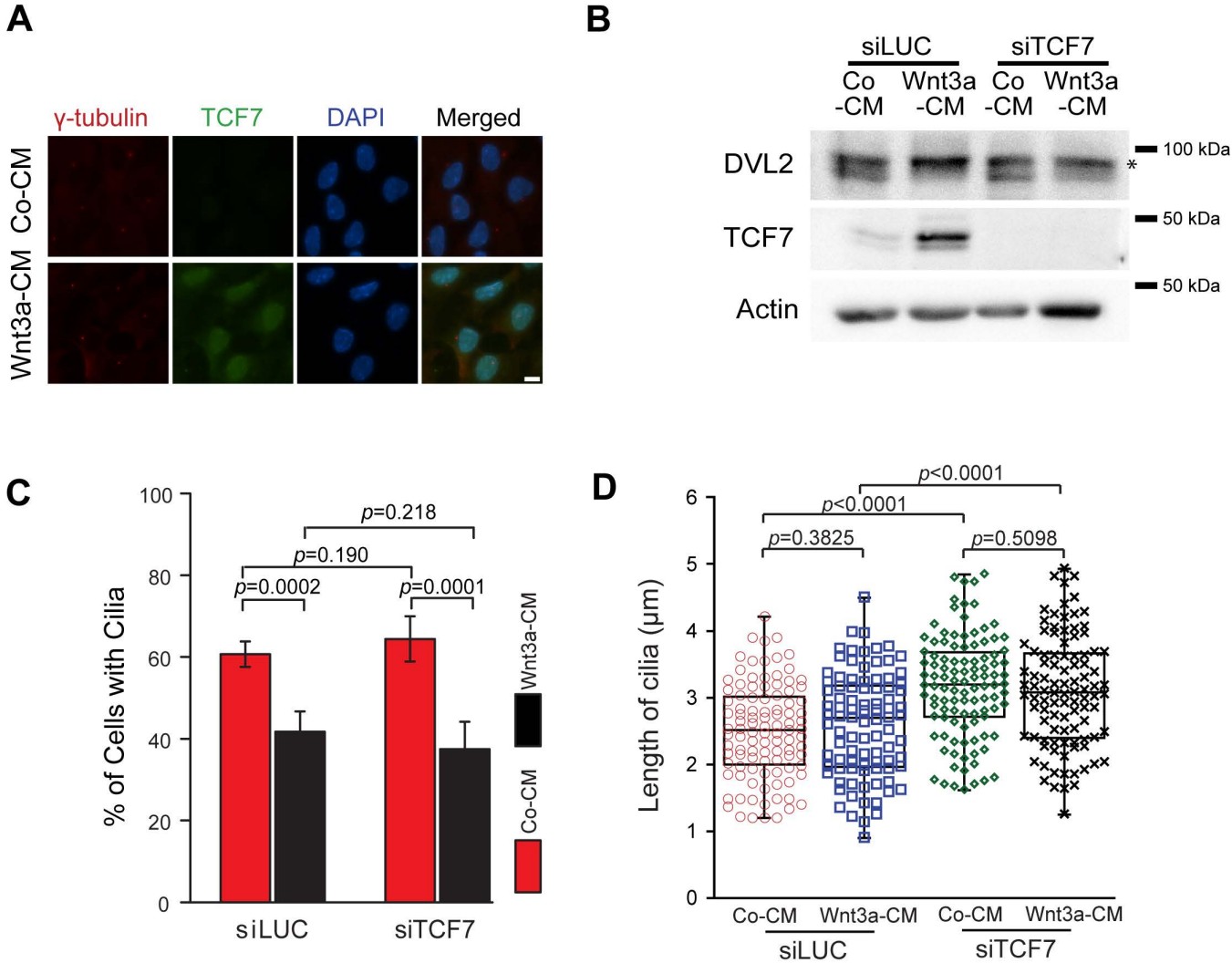

**Fig 2. Increasing baseline Wnt activity delays cilia formation independently of TCF7. (A)** Immunofluorescence images showing the localization of TCF7 (green) in Co-CM and Wnt3a-CM treated RPE1 cells as depicted in 1C. γ-tubulin (red) and DAPI (blue) serve as markers for centrosomes and nuclei, respectively. Scale bar: 5 μm. **(B, C)** RPE1 cells incubated with control (siLUC) or TCF7 siRNAs were treated with Co-CM or Wnt3a-CM as depicted in 1C and analyzed 16 h after serum starvation. (B) Western blot analysis of saponin-lysed RPE1 cells using DVL2 and TCF7 antibodies. Actin served as a loading control. The asterisk marks the phosphorylated form of DVL2. (C) Quantification of the percentage of ciliated cells based on γ-tubulin and ARL13b staining. The bar graph indicates the mean ± S.D. of three independent experiments. siLUC + Co-CM, $n = 338$; siLUC + Wnt3a-CM, $n = 325$; siTCF7 + Co-CM, $n = 336$; siTCF7 + Wnt3a-CM, $n = 322$. **(D)** Quantification of ciliary length from (C). The box/dot plots show quantification of ciliary length from three independent experiments. siLUC + Co-CM, $n = 104$; siLUC + Wnt3a-CM, $n = 93$; siTCF7 + Co-CM, $n = 107$; siTCF7 + Wnt3a-CM, $n = 100$. P values are based on Student $t$ test. The data underlying the graphs and blots in this figure can be found in the S1 Data and S1 Raw Images files.

(Figs 3A and S4A) and the accumulation of RAB8a around centrosomes (Fig 3B). The few cells that were still able to form cilia upon GSK3 inhibition had significantly longer cilia in comparison to wild-type cells (Fig 3C), supporting the previously reported influence of Wnt activation on cilia length [36]. We next asked whether Wnt3a-CM treatment phenocopied GSK3 inactivation in impairing RAB8a-vesicular trafficking to the centrosome. To investigate this, we used an established RPE1 cell line stably expressing GFP-RAB8a [10]. Whereas GFP-RAB8a accumulated around centrosomes shortly after serum starvation in control cells, the number of GFP-RAB8a-positive centrosomes was significantly reduced

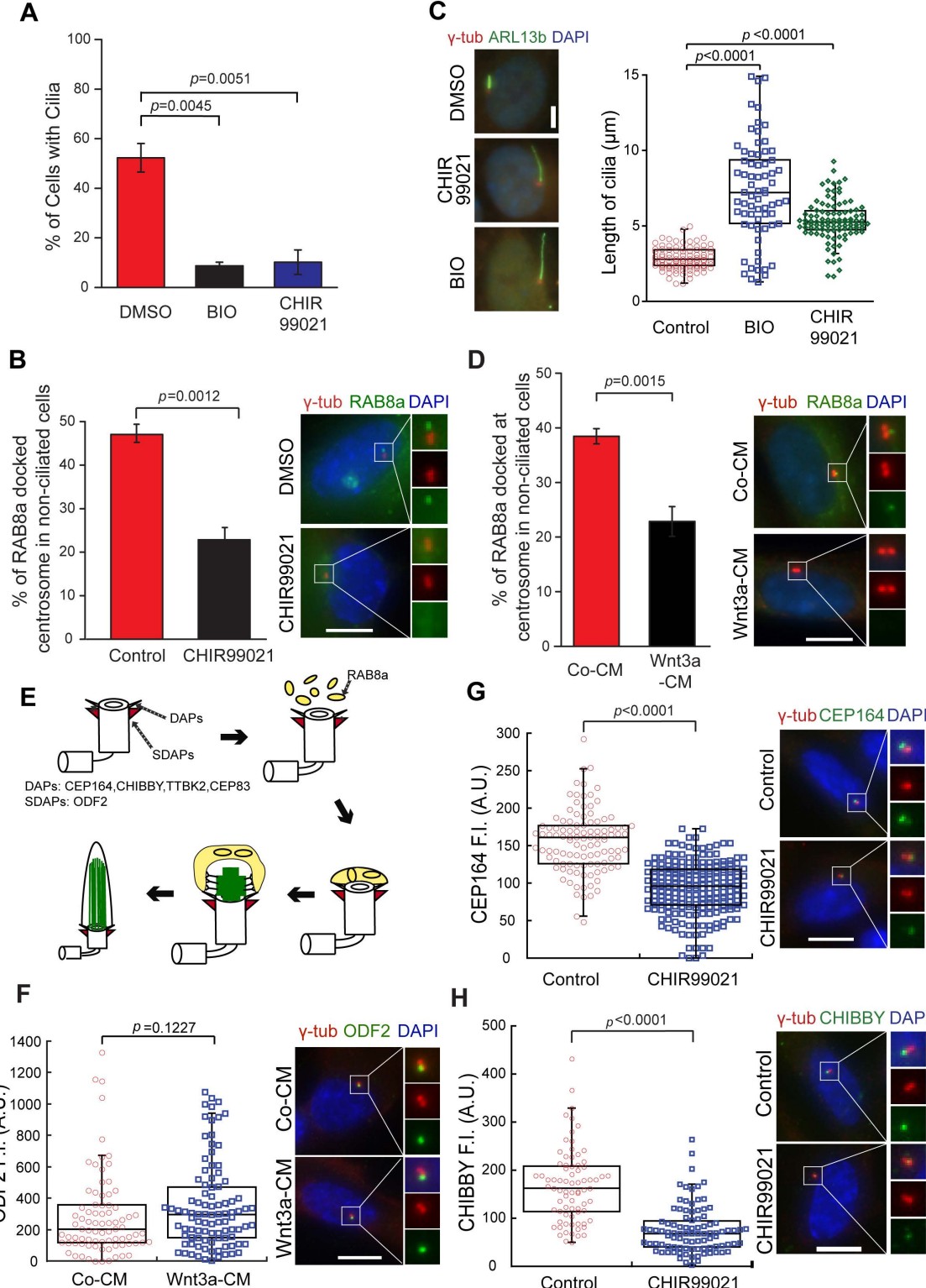

**Fig 3. Wnt signaling decreases the recruitment of RAB8a to basal bodies. (A)** RPE1 cells were serum starved for 16 h in the presence of solvent control (DMSO), BIO or CHIR99021. Cells were processed for immunofluorescence analysis using γ-tubulin and ARL13b antibodies. DNA was

stained with DAPI. The bar graph shows the percentage of ciliated cells (mean ± S.D.) of three independent experiments. DMSO, $n = 449$; BIO, $n = 497$; CHIR99021, $n = 391$. **(B)** RPE1 cells stably expressing GFP-RAB8a were treated with CHIR99021 as described in (A). GFP-RAB8a was detected by direct fluorescence. The graph shows the percentage of non-ciliated cells with GFP-RAB8a docked at centrosomes (stained with γ-tubulin antibodies). DNA was stained with DAPI. The mean ± S.D. of three independent experiments is shown. Representative images with enlarged centrosomal areas (as indicated) show merged γ-tubulin and GFP-RAB8a (top panel), γ-tubulin (red, middle panel) and GFP-RAB8a (green, lower panel) in control (DMSO) and CHIR99021-treated cells. DMSO, $n = 561$; CHIR99021, $n = 396$. Scale bar: 5 µm. **(C)** Quantification of ciliary length from (A). Representative images show basal body (γ-tubulin, red) and ciliary membrane (ARLB13b, green) merged signals. The box/dot plots show quantification of ciliary length from three independent experiments. Control, $n = 99$; BIO, $n = 68$; CHIR99021, $n = 99$. Scale bar: 5 µm. **(D)** RPE1 cells stably expressing GFP-RAB8a were treated with Co-CM and Wnt3a-CM as depicted in 1C and fixed for GFP-RAB8a localization analysis as described in (B). Quantifications and representative images are shown. Enlarged images show merged γ-tubulin (red) and GFP-RAB8a (green) signals (top panel). DNA was stained with DAPI. Co-CM, $n = 216$; Wnt3a-CM, $n = 306$. Scale bar: 5 µm. **(E)** Cartoons representing daughter and mother centrioles with distal (DAs) and subdistal (SDAs) appendages and components analyzed. RAB8a-positive vesicles (yellow) dock at the mother centriole at initial steps of ciliogenesis prior to extension of the axoneme (green). Created using Adobe Illustrator CS3. **(F)** RPE1 cells were treated as depicted in 1C and stained for ODF2 (SDA component, green) and γ-tubulin (red). The box/dot plots show quantification of fluorescence intensities of ODF2 at the centrosomes from three independent experiments. Representative images are shown on the right. DNA was stained with DAPI (blue). Co-CM, $n = 84$; Wnt3a-CM, $n = 97$. Scale bar: 10 µm. **(G, H)** RPE1 cells were treated with DMSO control or CHIR99021 and serum starved for 16 h prior to immunofluorescence analysis for CEP164 and CHIBBY. The box/dot plots show quantification of fluorescence intensities of CEP164 (G) and CHIBBY (H) at the centrosomes from three independent experiments. Representative images with enlargements of the centrosomal area are shown on the right. DNA was stained with DAPI (blue). **(G)** Control, $n = 110$; CHIR99021, $n = 111$. **(H)** Control, $n = 91$; CHIR99021, $n = 98$. Scale bar: 10 µm. $P$ values are based on Student $t$ test. A.U., arbitrary units. The data underlying the graphs in this figure can be found in the S1 Data file.

in Wnt3a-CM-treated cells (Fig 3D). Since RAB8a is transported to the centrosome in a microtubule-dependent manner [47,48], we asked whether the microtubule-dependent trafficking of proteins to the centrosome was generally disturbed upon Wnt activation. However, this was not the case, as the levels of the centriolar satellite protein PCM1, which accumulates around centrosomes via microtubule-based transport [15], did not differ upon Wnt3a treatment (S4B Fig).

RAB8a-vesicle docking at mother centrioles requires distal appendage proteins (Fig 3E) [10,11,49]. To investigate whether Wnt activation negatively influenced appendage formation, we compared the levels of distal appendage proteins at the basal body in Co-CM and Wnt3a-CM-treated cells. No significant decrease in protein levels was observed for the distal appendage components CEP164, TTBK2, or CHIBBY, whereas CEP83 levels were significantly increased (S4C–S4F Fig). The level of the subdistal component ODF2 at centrioles remained unchanged following Wnt3a-CM treatment (Fig 3F) or GSK3-inhibition (S4G Fig). However, GSK3-inhibited RPE1 cells showed decreased levels of CEP164 and CHIBBY (Fig 3G and 3H) and increased levels of CEP83 (S4H Fig) compared to control cells. The decrease in CEP164 and CHIBBY also occurred in GSK3-inhibited cycling cells not subjected to serum starvation (S5A and S5B Fig).

These data raised the question of whether GSK3 kinase plays a role in appendage integrity or assembly. To further explore this possibility, we performed electron microscopy of GSK3-inhibited cells upon serum starvation. In both control and CHIR99021-treated cells, subdistal and distal appendages were present at the mother centriole even though cilia were absent in CHIR99021-treated centrioles (S5C Fig). Similar results were obtained upon GSK3-inhibition using BIO (S5C Fig).

To analyze the spatial organization of CEP164 and CHIBBY at mother centrioles, we employed ultra-expansion microscopy (U-ExM). Side views of centrioles (detected by acetylated tubulin) revealed that CEP164 and CHIBBY decorated the distal end of the centriole in both control and GSK3-inhibited conditions (S5D and S5E Fig). Top views of centrioles showed the characteristic 9-fold assemblies of CEP164 in both control and GSK3-inhibited conditions (S5D Fig). Similarly, no notable difference was observed for CHIBBY localization in top views, where CHIBBY appeared as dot-like ring-shaped structures close to the centriolar wall in both conditions (S5E Fig). Therefore, we conclude that GSK3 inhibition decreases the levels of CEP164 and CHIBBY without affecting the ultrastructure of centrioles or the spatial organization of these proteins.

Collectively, the data indicate that early steps of ciliogenesis are delayed under high Wnt activity.

## Wnt signaling activation impairs the removal of the cilia inhibitory proteins CP110 and OFD1

An important step that occurs before ciliary membrane and axoneme extension is the downregulation of cilia-inhibitory components, including the protein complex CP110-CEP97 and the cytoplasmic pool of the protein OFD1 [4,6,19]. To test whether CP110-CEP97 or OFD1 regulation was impaired by Wnt overactivation, we first examined the centriolar pool of CP110 as a representative of the CP110-CEP97 complex. CP110 is present at both mother and daughter centrioles, but it is removed from the mother centriole prior to axoneme extension [6]. In Co-CM samples, CP110 was absent from the majority of the mother centrioles, labelled by the subdistal appendage component ODF2 (Fig 4A). In contrast, the percentage of mother centrioles associated with CP110 was significantly higher after Wnt3a-CM treatment (Fig 4A), indicating that cilia formation is impaired prior to CP110-CEP97 removal.

Next, we examined the centriolar satellite pool of OFD1, which is present at the cytoplasm and is degraded during early stages of ciliation [19], and found that it was increased in Wnt3a-CM cells compared to controls (Fig 4B and 4C). These data suggest that satellite-bound OFD1 is not efficiently removed in Wnt-stimulated cells, preventing axoneme assembly.

## Removal of OFD1 but not CP110 restores ciliogenesis in Wnt-activated cells

We reasoned that if the inhibition of cilia formation upon Wnt activation is caused by the persistent presence of CP110-CEP97 or OFD1, inducing their degradation should restore ciliogenesis in Wnt-stimulated cells. To test this, we used siRNA to down-regulate CP110 or OFD1 prior to ciliogenesis induction and Wnt3a-CM treatment. The depletion of CP110 did not change the effect of Wnt3a on ciliogenesis (Figs 4D, 4E, and S6). In contrast, OFD1 depletion significantly increased the percentage of ciliated cells in Wnt3a-CM-treated samples (Fig 4F and 4G). The knockdown of OFD1 also increased ciliogenesis in Co-CM treated cells (Fig 4F), which is consistent with data showing that removal of OFD1 promotes ciliogenesis in mouse embryonic fibroblasts [19]. Importantly, Wnt activation was not impaired after OFD1 depletion, as determined by phosphorylation of LRP6 and increased TCF7 protein levels (Fig 4G). Interestingly, while OFD1 depletion under Co-CM conditions resulted in an approximately 1.5-fold increase in cilia formation, this increase was amplified to 2.8-fold when the Wnt pathway was activated with Wnt3a-CM. Moreover, the overall cilia formation observed in response to OFD1 depletion remained similar across conditions. These findings suggest that the failure to remove OFD1 is a key factor contributing to reduced cilia formation during Wnt pathway hyperactivation.

## Inhibition of mTOR signaling rescues cilia formation in Wnt-activated cells

Given that Wnt3a functioned independently of TCF7 in ciliogenesis, we next asked how activated Wnt/LRP6 signaling impairs cilia formation by exploring Wnt/STOP and Wnt/mTOR cascades [34,50]. Of note, mTOR activity reduces autophagy [51,52], which is related to ciliogenesis through OFD1 downregulation [19]. To estimate whether the autophagic flux was indeed decreased in cells treated with Wnt3a-CM, we performed imaging analysis using RPE1 cells stably expressing the microtubule-associated protein light chain 3 (LC3) fused to mCherry and GFP (mCherry-GFP-LC3 reporter) [53,54]. The mCherry-GFP-LC3 fusion protein gives a green and red signal at autophagosomes, but only a red signal is detected at autolysosomes, due to the instability of GFP at acidic pH [53,54]. To quantify the number of autophagosomes, we counted the yellow foci per cell, representing co-localized mCherry and GFP signals (mCherry+GFP+ puncta). In comparison to Co-CM, the number of mCherry+GFP+ puncta in Wnt3a-CM-treated cells was significantly decreased (Fig 5A and 5B). This indicates a reduction in autophagosome formation following Wnt overactivation in serum-starved cells. We next determined mTOR activity in our conditions by looking at the phosphorylation levels of the mTOR substrate, the ribosomal protein S6 kinase (S6K), using a phospho-specific S6K(pT389) antibody. Phosphorylation of S6K was markedly increased in Wnt3a-CM (Fig 5C, lane 3) in comparison to control (Fig 5C, lane 1). This effect was due to mTOR activity, as S6K phosphorylation was blocked by the mTOR inhibitor, rapamycin (Fig 5C, lanes 2 and 4).

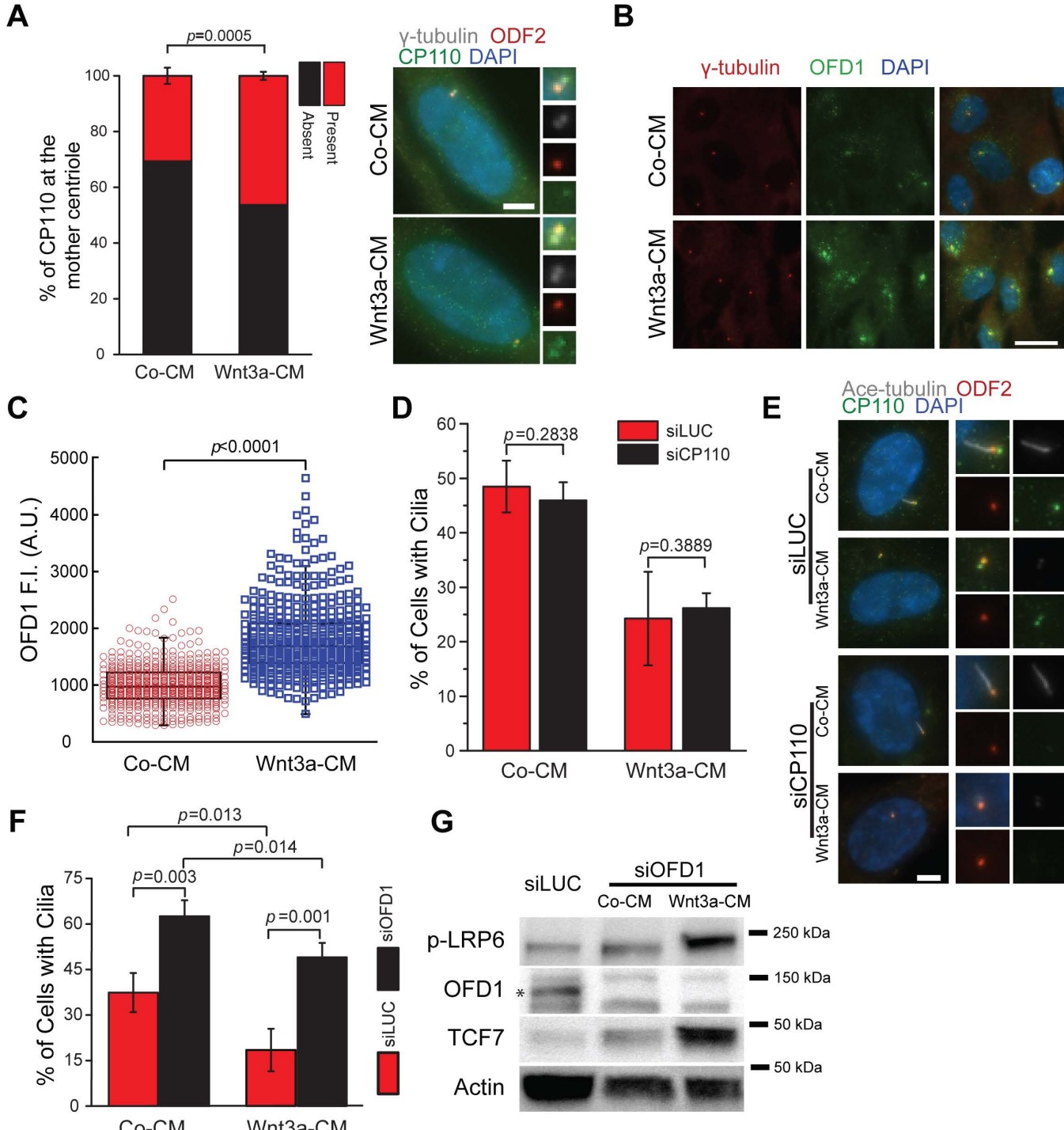

**Fig 4. Wnt signaling activation impairs the removal of the cilia inhibitory proteins CP110 and OFD1. (A)** RPE1 cells were treated as depicted in 1C and stained for CP110 (green), ODF2 (red) and γ-tubulin (gray). The graph shows the percentage of mother centrioles (marked by ODF2) with or without CP110. Mean± S.D. of three independent experiments are shown. Representative images with enlargements of the centrosomal area as depicted are shown on the right. Co-CM, $n = 455$; Wnt3a-CM, $n = 390$. DNA was stained with DAPI (blue). Scale bar: 5 μm. **(B)** RPE1 cells were treated as depicted in 1C and stained for OFD1 (green) and γ-tubulin (red) after 16 h of serum starvation. Representative images show OFD1 localization in Co-CM or Wnt3a-CM treated cells. DNA was stained with DAPI (blue). Scale bar: 20 μm. **(C)** Quantification of the cytoplasmic centriolar satellite OFD1

signal of (B) in arbitrary units. Co-CM, *n*=480; Wnt3a-CM, *n*=430. **(D)** RPE1 cells incubated with control (siLUC) or CP110 siRNAs were treated with Co-CM or Wnt3a-CM as depicted in 1C and analyzed by immunofluorescence 16 h after serum starvation. The bar graph indicates the percentage of ciliated cells (mean ± S.D.) from three independent experiments. siLUC + Co-CM, *n*=436; siLUC + Wnt3a-CM, *n*=393; siCP110 + Co-CM, *n*=235; siCP110 + Wnt3a-CM, *n*=338. **(E)** Representative images of (D). Cells were stained with acetylated tubulin (Ace-tubulin, gray), ODF2 (mother centriole marker, red), CP110 (green). Enlargements are shown on the right. Scale bar: 5 μm. **(F)** RPE1 cells incubated with control (siLUC) or OFD1 siRNAs were treated with Co-CM or Wnt3a-CM as depicted in 1C and analyzed by immunofluorescence 16 h after serum starvation. The bar graph shows the percentage of ciliated cells (mean ± S.D.) from three independent experiments. siLUC + Co-CM, *n*=690; siLUC + Wnt3a-CM, *n*=677; siOFD1 + Co-CM, *n*=521; siOFD1 + Wnt3a-CM, *n*=509. **(G)** Western blot analysis of (F) showing the levels of phospho-LRP6 (p-LRP6), OFD1 (marked by an asterisk) and TCF7 in the indicated samples serum starved for 16 h. Actin served as a loading control. *P* values are based on Student *t* test. The data underlying the graphs and blots in this figure can be found in the S1 Data and S1 Raw Images files.

A way to distinguish between Wnt/STOP and Wnt/mTOR signaling is to assess whether the effects are rapamycin-dependent [30]. Blocking mTOR with rapamycin strongly increased ciliation in Wnt3a-CM-treated cells, but not in control cells (Fig 5D). mTOR inhibition also rescued ciliogenesis when GSK3 kinase activity was blocked by CHIR99021 (Fig 5E). Taken together, these data suggest that increased mTOR activity contributes to cilia loss under high Wnt baseline activation.

## Discussion

Our findings indicate that balanced Wnt activity is crucial for proper ciliogenesis, as both excessive and insufficient Wnt signaling result in defects in cilia formation (Fig 5F). Consistently, the depletion of Wnt pathway components LRP5/6 or β-catenin reduces ciliogenesis in RPE1 cells, supporting the previously proposed positive role of Wnt signaling in cilia bio-genesis [35–37]. However, the conclusion that basal Wnt activity contributes to cilia formation cannot be generalized, as depletion of LRP5/6 or Axin had no effect on ciliogenesis in other studies [37,38]. These differences may arise from growth conditions or intrinsic variations between cell types. For instance, HEK293 cells have a much lower rate of ciliation com-pared to RPE1 cells, a behavior that may interfere with the effect of basal Wnt activity on cilia initiation dynamics. Genetic variations between cell lines of similar origin may also play a role, as the depletion of LRP5/6 impaired ciliogenesis in the HEK293T but not in its parental HEK293 cell line [37]. These observations underscore the need for further detailed analy-sis, considering not only the cellular context but also the genetic background of the cell lines being analyzed.

Our data suggest that elevated baseline Wnt activity reduces, but does not entirely block, ciliogenesis in RPE1 cells. In our protocol, Wnt agonists were applied to cycling cells 6 h prior to inducing ciliogenesis through serum starvation. This brief, acute treatment effectively enhanced Wnt activity while preserving the cells' ability to arrest in the G1/G0 phase of the cell cycle, a critical prerequisite for cilia biogenesis. The same treatment regimen also decreased cilia formation in murine NIH3T3 and IMCD3 cells, ruling out an RPE1-specific behavior. Previous reports concluded that Wnt activation promotes or has no effect on cilia formation in RPE1 cells [36,38], although Bernatik and colleagues [38] reported a mild decrease in ciliation, which aligns with our observations. We reason that differences in cell type, confluency, or timing of Wnt activation relative to cilia initiation may account for these discrepancies. Early steps of ciliation involve the docking, fusion, and extension of ciliary vesicles at the mother centriole. Based on early vesicle-associated markers, cilia initiation has been shown to occur shortly (within 15–30 min) after serum starvation, although cilia extension dynamics may vary depending on the cell type and experimental condition [55,56]. In previous studies, Wnt ligands were added 6 h or even 48 h after serum withdrawal [36,38]. It is thus possible that canonical Wnt signaling activation is unable to interfere with ciliation once ciliary vesicle establishment and axoneme extension have begun at the mother centriole.

Our data indicate that early steps of cilia biogenesis are impaired when cells with an increased basal Wnt activity are triggered to ciliate through serum withdrawal. At this early stage, the mother centriole capping CP110-Cep97 com-plex blocks axoneme extension, and its removal not only allows axoneme microtubule elongation but also facilitates the recruitment of Rab8-positive ciliary vesicles to basal bodies for cilia membrane elongation [4]. In serum-starved cells

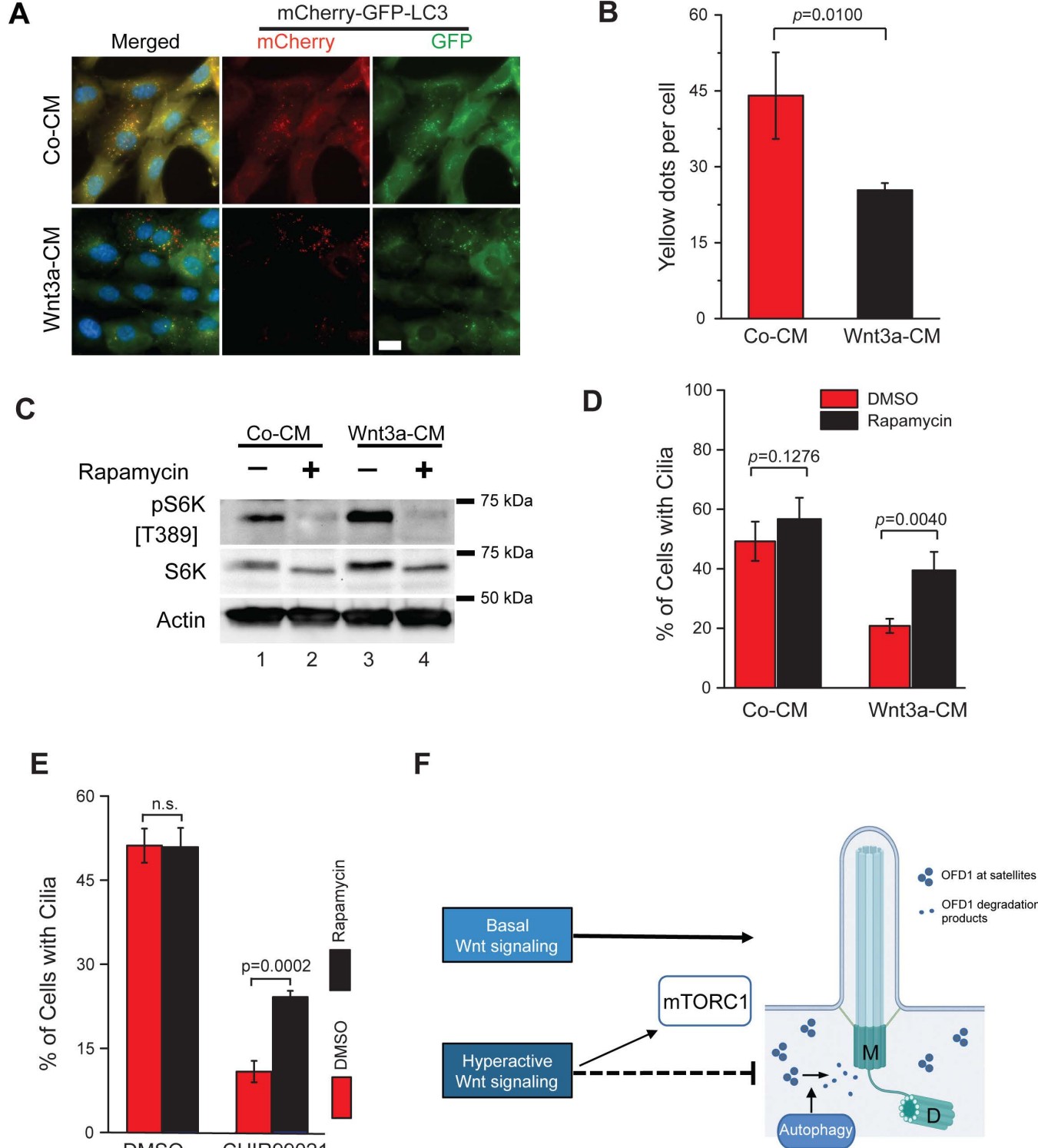

**Fig 5. Inhibition of mTOR signaling rescues cilia formation in Wnt activated cells. (A, B)** RPE1 cells stably expressing mCherry-GFP-LC3 were treated with Co-CM and Wnt3a-CM as depicted in 1C and fixed 16 h after serum starvation for direct fluorescence analysis of mCherry and GFP signals. Representative images (A) and quantification of the number of GFP+mCherry+ foci (yellow dots, autophagosomes) (B) from three independent experiments are shown. Co-CM, $n = 123$; Wnt3a-CM, $n = 113$. Scale bar: 10 μm. **(C)** RPE1 cells were treated as depicted in 1C in the presence or absence of

rapamycin and analyzed by western blot 16 h after serum starvation using phospho-S6K (T389) and S6K antibodies. Actin served as a loading control. DMSO was used as a solvent control. **(D)** RPE1 cells were treated as depicted in 1C in the presence of solvent control (DMSO) or rapamycin and analyzed by immunofluorescence 16 h after serum starvation. The bar graph shows the percentage of ciliated cells from three independent experiments. Co-CM+DMSO, $n = 394$; Co-CM+Rapamycin, $n = 361$; Wnt3a-CM+DMSO, $n = 468$; Wnt3a-CM+Rapamycin, $n = 372$. **(E)** RPE1 cells were treated with solvent control (DMSO) or CHIR99021 in the presence or absence of rapamycin as indicated and serum starved for 16 h. The percentage of ciliated cells was quantified based on ARL13b (cilia membrane marker) and γ-tubulin (basal body marker). The graph shows the mean ± S.D. from three independent experiments. DMSO, $n = 484$; DMSO+rapamycin, $n = 588$; CHIR99021 + DMSO, $n = 478$; CHIR99021 + Rapamycin, $n = 476$. $P$ values are based on Student $t$ test. **(F)** Model of how Wnt signaling affects ciliogenesis. Basal Wnt signaling promotes cilia formation, whereas Wnt hyperactivation prior to ciliogenesis delays this process by increasing mTORC1 activity and impairing the removal of OFD1 from centriolar satellites. M, mother centriole; D, daughter centriole. The cilia illustration was created using ChatGPT and Adobe Illustrator CS3. The data underlying the graphs and blots in this figure can be found in the S1 Data and S1 Raw Images files.

with enhanced Wnt basal activity, induced by either Wnt3a addition or GSK3 inhibition, a significantly higher percentage of cells retained CP110-Cep97 at mother centrioles, explaining the reduced accumulation of Rab8-positive vesicles at mother centrioles. GSK3-inhibition, but not Wnt3a-treatment, decreased the levels of the distal appendages proteins CEP164 and CHIBBY at mother centrioles. The reason for this difference remains unclear, suggesting a more complex role for Wnt signaling in ciliogenesis. Wnt activation effectively inhibits GSK3 from phosphorylating axin and β-catenin, but not additional GSK3 substrates that are unrelated to the destruction complex [57]. It is thus possible that an active pool of GSK3 remains in Wnt-stimulated cells, regulating the levels of CEP164 and CHIBBY at distal appendages.

We found that Wnt hyperactivation prior to serum starvation affects the satellite pool of OFD1. This pool is typically removed through selective autophagy, facilitating the recruitment of components like the Bardet–Biedl Syndrome protein BBS4, which promote axoneme extension at the basal bodies [19]. We propose that a defective autophagy-dependent removal of OFD1 at satellites is one key underlying mechanism delaying ciliogenesis in cells with high baseline Wnt activity. This notion is supported by our findings showing reduced autophagosome formation in cells with hyperactive Wnt, higher OFD1 levels at centriolar satellites, and increased ciliogenesis upon ectopic depletion of OFD1 in cells with hyperactive Wnt signaling.

We reason that the decreased autophagy under high baseline Wnt activity is related to mTOR signaling, which inhibits autophagy through phosphorylation of multiple autophagy-related components involved in both early and late stages of autophagosome maturation [52]. The activation of Wnt signaling through LRP5/6 and Wnt3a ligand has been shown to stimulate mTOR activity in a GSK3-dependent manner, independently of β-catenin [34]. Our data show that, compared to serum-starved cells with basal Wnt activity, serum-starved cells treated with Wnt3a exhibited higher mTOR activity, as indicated by increased phosphorylation of S6K. Under this condition, the phosphorylation of S6K was sensitive to rapamycin, in agreement with the function of Wnt signaling in stimulating mTORC1 activity and S6K phosphorylation in a rapamycin-sensitive manner [34]. We thus postulate that when RPE1 cells with enhanced Wnt baseline activity are serum-starved, the decrease in GSK3 activity prior to serum starvation enhances mTOR activity, which in turn compromises OFD1-dependent autophagy removal and ciliogenesis. In agreement with this hypothesis, rapamycin treatment also rescued ciliogenesis in GSK3-inhibited cells. Interestingly, somatic hypermorphic mutations in mTOR are found in patients with neuronal development disorders related to focal malformation of cortical development (FMCD) [58]. In mouse models, hyperactive mTOR mutants result in loss of cilia due to impaired autophagy and OFD1 removal [59]. Similar to our results in RPE1 cells, mTOR inhibition as well as ectopic OFD1 removal restored ciliogenesis and neuronal migration in FMCD mouse models [59], highlighting the importance of this regulation in a physiological context.

Wnt signaling and cilia are both key regulators of developmental programs, with defects in cilia and dysregulated Wnt signaling frequently implicated in pathological conditions such as developmental disorders, organ dysfunction, and cancer [60]. Notably, Wnt hyperactivation in murine cerebral cortical progenitors with APC loss-of-function or β-catenin gain-of-function mutations reduces cilia formation and disrupts cortical development [39], highlighting the importance of optimal Wnt activity in cerebral cortical progenitor development. A critical question is how individual cells in the context

of a developing tissue fine-tune Wnt signaling activation to ensure that cilia biogenesis initiates at the mother centriole at the correct developmental time. We propose that Wnt activity operates within a critical window to either stimulate or inhibit ciliogenesis, involving cross-talk with the mTOR pathway. In turn, cilia contribute to Wnt pathway activation [61–63]. The Wnt-cilia relationship is further complicated by intricate interactions with other pathways, including mTOR, BMP, Notch, insulin, and Hippo, many of which both influence or are influenced by ciliogenesis [1,33,64,65]. These findings underscore the tightly regulated interplay between these pathways, fine-tuned during development and responsive to environmental cues. Exploring these complex, tissue-specific interactions will yield valuable insights into the diverse phenotypes of ciliopathies and Wnt-related diseases.

## Materials and methods

### Cell culture conditions and treatments

RPE1 cells were cultured in DMEM/F12 (Sigma Aldrich) supplemented with 10% fetal bovine serum (FBS, Biochrom), 2 mM L-glutamine (Thermo Fisher Scientific), and 0.348% NaHCO₃ (Sigma Aldrich). HEK293T cells were maintained in DMEM high-glucose medium supplemented with 10% FBS. NIH3T3 cells were grown in DMEM high glucose (Sigma Aldrich) with 10% newborn calf serum (NCS, PAN-Biotech). IMCD3 cells were cultured in DMEM/F12 supplemented with 10% FBS. All cell lines were maintained at 37°C in a 5% $CO_2$ atmosphere.

To induce cilia formation by serum starvation, $2 \times 10^4$–$2.5 \times 10^4$ RPE1 or NIH3T3 cells were seeded on glass coverslips in 24-well plates for 24 h and incubated in serum-free medium for the indicated time points in each experiment. For IMCD3 cells, $2.5 \times 10^4$ cells were seeded on glass coverslips in 24-well plates for 24 h and incubated in DMEM/F12 supplemented with 0.5% FBS for 24–48 h. Cell numbers were determined using the Luna automated cell counter (Logos Biosystems).

Cells were treated with Co-CM and Wnt3a-CM (1:10 dilution in the corresponding media). CHIR99021 (Sigma Aldrich) was used at a final concentration of 5 µM, BIO (Cayman Chemical) was used at a final concentration of 0.67 µM. Rapamycin (Sigma Aldrich) was used at a final concentration of 1 µM.

### Cell lines and transfection

RPE1 cells were transiently transfected by electroporation (NEPA21 Transfection System, Nepa Gene) or with FuGENE 6 (Promega), following the manufacturer's protocols. RPE1 cells stably expressing mCherry-GFP-LC3 were a kind gift of Jon Lane (University of Bristol, UK) [66]. RPE1 cells stably expressing the Wnt reporter 7xTGC and HEK293T cells stably expressing Fzd subtype-specific, next-generation surrogate (NGS) Wnt agonist were constructed by lentivirus transduction using the plasmids 7xTGC (addgene #24304) [41] and RKS-Fzd7/8 subtype NGS Wnt (addgene #159628) [67], respectively. Lentivirus production was carried out in HEK293T cells using PEI (polyethyleneimine 25000, Polysciences) as transfection reagent and the packing plasmids pMD2.G (addgene #12259) and psPAX2 (addgene #12260). RPE1 or HEK293T GFP-positive cells were FACS-sorted 15 days after lentiviral transduction. RPE1 cells were transfected with 40 µg of plasmids expressing *WNT1* and *DKK1* [68,69].

### Antibodies

Primary antibodies used in this study are listed in S1 Table. Secondary antibodies were purchased from Thermo-Fisher Scientific and included: goat anti-rabbit, anti-mouse, or anti-guinea pig IgGs conjugated to HRP or Alexa Fluor 488, 594, or 647 (1:500 dilution).

### Small-interfering RNAs (siRNAs)

Detailed information can be found in S2 Table. Transfections of siRNA were performed with Lipofectamine RNAiMAX reagent according to the manufacturer's instructions (Thermo Fischer Scientific). Briefly, $3 \times 10^4$ cells were seeded per

well in a 24-well plate and analyzed 48 h after the initial transfection. For western blotting analysis, $1.2 \times 10^5$ cells were reverse-transfected in one 6-well plate.

## Conditioned media and Wnt surrogate purification

Control and Wnt3a-conditioned media were produced as previously described [68]. Cells were incubated with Co- or Wnt3a-CM at 1:10 dilution. For surrogate Wnt agonist purification, HEK293T stably expressing the 6xHis-tagged surrogate Wnt agonist were seeded in one 15-cm dish and grown in DMEM/F12-high glucose (Sigma Aldrich) media supplemented with 10% FBS for 7 days before separating the cells and cell culture supernatant by centrifugation. The supernatant was maintained on ice. The cell pellet was resuspended in lysis buffer (50 mM $NaH_2PO_4$, 300 mM NaCl, 100 mM imidazole, pH 8.0) and sonicated until cells were lysed. The lysate was centrifuged at 10,000$g$ for 20–30 min at 4 °C to remove cell debris. The cleared lysate was collected, mixed with the cell culture supernatant, and incubated with 1 ml of Ni-NTA Agarose (50% slurry in lysis buffer) for 1–2 h on a rotation wheel at 4 °C. After 4 washes with washing buffer (50 mM $NaH_2PO_4$, 300 mM NaCl, 20 mM imidazole, pH 8.0), the recombinant protein was eluted in batch seven times with 0.5 ml of elution buffer (50 mM $NaH_2PO_4$, 300 mM NaCl, 250 mM imidazole, pH 8.0). The fractions were analyzed by SDS-PAGE and tested for activity in a TOPflash luciferase Wnt reporter assay, as previously described [68]. Active fractions were kept at −80 °C for long-term storage. Cells were incubated with 25 µg/ml of purified surrogate Wnt agonist.

## FACS analysis of DNA content

Cells were fixed with 70% ice-cold ethanol and incubated with staining solution composed of 50 µg/ml propidium iodide, 0.08% Triton X-100, 0.2 mg/ml RNase A, and 1 mM EDTA in phosphate-buffered saline (PBS). Cells were subjected to DNA content analysis on a BD FACSCanto flow cytometer (BD Biosciences).

## Indirect immunofluorescence and microscopy

Cells were fixed with pre-cooled methanol at −20 °C for 5 min. Cells expressing fluorescence fusion proteins were fixed with 3% paraformaldehyde solution in PBS for 3 min at room temperature followed by cold methanol fixation for another 4 min at −20°C. For acetylated-tubulin stainings, cells were placed on ice for 20 min before fixation. Cells were blocked with PBST (PBS containing 3% IgG-free BSA and 0.1% Triton X-100) for 30 min, incubated with primary antibodies in a wet and dark chamber for 1 h and with secondary antibodies for 45 min at room temperature. All antibodies were diluted in PBST. DNA was stained with 50 ng/ml 4′,6-diamidino-2-phenylindole (DAPI) added to the secondary antibody solution. Samples were mounted with Mowiol (EMD Millipore) before inspection. Images were acquired as Z-stacks using an automated Nikon Ti2 microscope coupled to a Scientific IRIS9 CMOS camera (Photometrics) and Nikon NIS-Elements software or with an automated Zeiss Axiovert Observer coupled to an AxioCam MRm CCD camera and Zeiss ZEN software.

For live cell imaging, RPE1 cells ($4 \times 10^4$) stably expressing the 7xTGC reporter were seeded in a glass-bottom round dish (Ibidi) in HEPES-buffered DMEM/F12 medium without phenol red (Thermo Fisher Scientifics) and containing 10% FBS, 1% L-glutamine, and 1% penicillin-streptomycin. After 24 h, the medium was exchanged to serum-free medium in the presence of Co-CM or Wnt3a-CM media. Images were acquired every 30 min for 24 h using a Nikon Ti-T2 microscope.

## Ultrastructure expansion microscopy (U-ExM)

U-ExM was performed following a protocol modified from Gambarotto and colleagues, 2019 [70]. Briefly, cells seeded on coverlips were washed once with PBS and pre-fixed for 5 min in pre-cooled methanol at −20 °C. After three washes with PBS, coverslips were incubated in 0.7% formaldehyde (Sigma Aldrich) and 1% acrylamide (Bio-Rad) in PBS for 4–5 h at 37 °C. For gelification, the coverslips were placed upside down on a 35 µl-drop of monomer solution in a humid chamber at −20°C for 5 min and 37 °C for 1 h. The monomer solution is composed of 21.1% sodium acrylate (AK

Scientific), 11.1% acrylamide, and 0.22% N,N′-methylenbisacrylamide (Roth) in PBS, freshly supplemented with 0.5% N,N,N′,N′-Tetramethylethylenediamine (TEMED) and 0.5% ammonium persulfate (APS). The coverlips-carrying gels were placed in denaturation buffer (5.71% SDS, 200 mM NaCl, 0.6% Tris, pH 9.0) at room temperature for 15 min under shaking. Gels were removed from coverslips and incubated under shaking in denaturation buffer at 95 °C for 30 min. Gel expansion was performed by incubation in 100 ml double-distilled water (ddH$_2$O) at room temperature under shaking for 1 h, changing the ddH$_2$O every 20 min. The gel size was measured and compared with the initial coverslip size for calculation of the expansion factor. The expansion factor for samples shown in S5D and S5E Fig was 4.0–4.2.

Prior to immune staining, gels were shrunk by three 10-min incubations in 100 ml PBS. Gels were incubated at 37 °C overnight with primary antibodies diluted in PBS containing 1% BSA. Gels were washed three times with 0.1% Tween20 in PBS, before 2.5 h incubation at 37 °C with secondary antibodies diluted in PBS/1% BSA containing 50 ng/ml DAPI. All incubations were performed under shaking. After three washes with 0.1% Tween20 in PBS, gels were expanded in ddH$_2$O as previously. For imaging, gels were placed on a glass bottom dish (MatTek, Cat. Nr. P35G-1.5-14-C) coated with poly-l-lysine solution (Sigma Aldrich).

Images were taken on a Nikon AX confocal microscope with a Plan Apo λ 100× Oil Ph3 DM oil Nikon objective coupled to the AX/AX R's base DUX-ST detector, and Nikon NIS-Elements AR software. Images were deconvolved using the Richardson-Lucy algorithm available in the 3D deconvolution module of Nikon NIS-Elements AR Analysis Software 5.30.06.

## Electron microscopy

RPE1 cells were seeded on coverslips and cultured till they reached a confluency of approximately 60%–80%. Cells were rinsed with PBS three times and pre-fixed with a mixture of 2.5% glutaraldehyde, 1.6% paraformaldehyde and 2% sucrose in 50 mM cacodylate buffer for 30 min at room temperature. After rinsing of the cells with cacodylate buffer, cells were post-fixed with 2% OsO4 for 45 min on ice in the dark. Cells were rinsed with dH$_2$O and incubated overnight at 4 °C in 0.5% uranyl acetate aqueous solution. On the following day coverslips were rinsed again with dH$_2$O and subsequently stepwise dehydrated with ethanol. Coverslips were immediately placed on capsules filled with Spurr-resin (Sigma Aldrich) and polymerized at 60 °C for approximately two days. In resin-embedded cells were serial-sectioned using a Reichert Ultracut S Microtome (Leica Instruments, Vienna, Austria) to a thickness of 80 nm. Post-staining was performed with UranyLess aqueous solution (Delta Microscopes, France) and lead citrate. Serial sections were imaged at a Jeol JE-1400 (Jeol, Tokyo, Japan) microscope, operating at 80 kV and equipped with a 4k × 4k digital camera (F416, TVIPS, Gauting, Germany). Micrographs were adjusted in brightness and contrast applying ImageJ software.

## Imaging processing and analysis

Raw images were imported to Fiji image software [71] for analysis and exported to Adobe Photoshop and Illustrator CS3 for panel arrangements. No manipulations were performed other than brightness, contrast, and color balance adjustments. For quantifications of signal intensity around centrosomes/basal bodies, fluorescence intensity was measured in Fiji using maximum projection of images. For this, two areas around the centrosome of 7 and 10 square pixels were measured for centrosome signal intensity and background correction, respectively. Centriolar satellite proteins were quantified with CellProfiler [72] using γ-tubulin to segment the centrosome signal. Measurement of cilia length was performed using the Measure Plugin in Fiji. Data are shown as mean ± standard deviation (S.D.), as indicated in the figure legends. Where indicated, Student t test (two groups) was calculated using KaleidaGraph (Synergy Software). Outliers were not excluded. The models and schemes were created with ChatGPT and modified in Adobe Illustrator CS3.

## Western blotting

For Western blotting, to extract cytoplasmic β-Catenin and phosphorylated proteins, cells were lysed in either RIPA buffer (supplemented with 1× protease phosphatase inhibitor cocktail; Roche) or cytoplasmic lysis buffer (PBS supplemented

with 0.05% saponin, 10 mM β-mercaptoethanol, 2 mM EDTA), both containing protease and phosphatase inhibitor cocktail. For other proteins, cells were lysed in 8 M Urea in 10 mM Tris-HCl pH 8 containing benzonase (Sigma, 1:1000 dilution of 250 units/µl stock). Samples were incubated at room temperature for 10 min. The protein concentration was measured by Bradford reagent (Sigma Aldrich) accordingly to manufacturer's protocol. Samples were heated up to 65 °C for 5 min before loading on SDS-PAGE gels. Proteins were transferred onto nitrocellulose membranes using semi-dry blotting. To detect proteins larger than 120 kDa by western blot, a wet immunoblot on a PVDF membrane (GE Healthcare Amersham) was performed in borate buffer (1.25 g/l boric acid, 0.3725 g/l EDTA, pH 8.8). Blotting was performed at 350 mA for 3 h at 4 °C. Membranes were blocked with 5% milk or 5% BSA in PBST for 30 min. The membrane was incubated with primary antibodies at 4 °C overnight. Horseradish peroxidase (HRP)-conjugated secondary antibodies were incubated with the membrane for 1 h at room temperature. Proteins were visualized by enhanced chemiluminescence (ECL; Pierce; Thermo Fisher Scientific) using a gel documentation system (INTAS Imager or Cytiva Amersham ImageQuant800).

## Supporting information

**S1 Fig. Disruption of baseline Wnt activity impairs cilia formation in RPE1 cells. (A)** Western blot analysis of LRP6 in control (siLUC) and LRP6-depleted (siLRP6) RPE1 cells, serum starved for 24 h. Actin served as a loading control. **(B)** Western blot analysis of β-catenin in control (siLUC) and β-catenin-depleted (siCTNNB1) RPE1 cells, serum starved for 24 h. Actin served as a loading control. **(C)** Quantification of ciliary length from (A). The box/dot plots show quantification of ciliary length from three independent experiments. siLUC, $n=180$; siLRP6, $n=87$. **(D)** Quantification of ciliary length from (B). The box/dot plots show quantification of ciliary length from three independent experiments. siLUC, $n=160$; siCTNNB1, $n=107$. $P$ values are based on Student $t$ test. The data underlying the graphs and blots in this figure can be found in the S2 Data and S1 Raw Images files. (S1_Fig.TIF)

**S2 Fig. Wnt/β-catenin signaling hyperactivation delays primary cilia formation in RPE1, NIH3T3, and IMCD3 cells. (A)** Images of RPE1 cells expressing the 7xTGC Wnt reporter construct treated with Wnt3a-CM for the indicated time points. The increase in GFP signal reflects Wnt activation. **(B)** Western blot analysis of saponin lysed RPE1 cells treated with Co-CM or Wnt3a-CM (as depicted in Fig 1C) and serum starved for 48 h. Actin served as a loading control. **(C)** Experimental setup and quantification of ciliation of RPE1 cells treated with Co-CM and Wnt3a-CM at the indicated times after serum starvation. The bar graph indicates the mean ± S.D. from three independent experiments. At least 150 cells were counted per sample and time point. **(D)** Pie chart showing the percentage of RPE1 cells in G1, S, and G2/M phases of the cell cycle as determined by FACS-based DNA content analysis after Co-CM and Wnt3a-CM treatment (as depicted in C) and 16 h of serum starvation. **(E)** Quantification of ciliation in NIH3T3 cells treated with Co-CM ($n=461$) and Wnt3a-CM ($n=493$) and serum starved for 16 h. The bar graph indicates the mean ± S.D. from three independent experiments. **(F)** Quantification of ciliation in IMCD3 cells treated with Co-CM or Wnt3a-CM and serum starved for 24 and 48 h. The bar graph indicates the mean ± S.D. from three independent experiments. **(G)** Relative luciferase activity in HEK293T cells treated either with Co-CM or Wnt3a-CM at the indicated dilutions or with purified recombinant surrogate Wnt agonist (concentrations varying between 1 and 10 µg/ml as indicated). Reactions were done in the absence (black bars) or presence (red bars) of R-Spondin to enhance Wnt signaling. DKK1 (Wnt inhibitor) was used as a control. Mean ± S.D. from three independent experiments are shown. **(H)** Images of RPE1 7xTGC cells treated for 6 h with buffer (control) or 10 µg/ml of surrogate Wnt agonist. The increase in GFP signal reflects Wnt activation. Scale bar: 10 µm. $P$ values are based on Student $t$ test. The data underlying the graphs and blots in this figure can be found in the S2 Data and S1 Raw Images files. (S2_Fig.TIF)

**S3 Fig. Determination of cilia length upon Wnt/β-catenin signaling activation. (A)** Quantification of ciliary length of RPE1 cells treated with serum-free medium (control), Co-CM and Wnt3a-CM and serum starved for 16 h. The box/dot plots show quantification of ciliary length from three independent experiments. Control, $n=90$; Co-CM, $n=95$; Wnt3a-CM,

*n* = 47. **(B)** Quantification of ciliary length of RPE1 cells treated with buffer control (*n* = 95) or purified surrogate Wnt agonist (*n* = 175) after 16 h of serum starvation. The box/dot plots show quantification of ciliary length from three independent experiments. **(C)** Quantification of Fig 2B showing the relative signal intensity of p-DVL2 to actin in siLUC and siTCF7 Wnt3a-CM treated samples. The bar graph indicates the mean ± S.D. of three independent experiments. *P* values are based on Student *t* test. The data underlying the graphs in this figure can be found in the S2 Data file.
(S3_Fig.TIF)

**S4 Fig. Analysis of PCM1 and appendages upon Wnt signaling activation. (A)** Western blot analysis of saponin lysed RPE1 cells shows the levels of cytoplasmic β-catenin after 16 h of serum starvation and treatment with DMSO, BIO or CHIR99021. Actin served as a loading control. **(B)** Representative images and quantification of PCM1 fluorescence intensity at the centrosomal area (in arbitrary units) in RPE1 cells treated with Co-CM and Wnt3a-CM as depicted in S2C and serum-starved for 16 h. γ-tubulin (red) and DAPI (blue) served as markers for centrosomes and nuclei, respectively. Magnifications of the centrosomal area are shown on the right as indicated. Co-CM, *n* = 214; Wnt3a-CM, *n* = 187. Scale bar: 10 µm. **(C–F)** RPE1 cells were treated with Co-CM and Wnt3a-CM and serum starved for 16 h as depicted in Fig 1C. Cells were stained for CEP164 (C), TTBK2 (D), CHIBBY (E), and CEP83 (F). γ-tubulin served as a centriolar marker. DNA was stained with DAPI. The graph shows the fluorescence intensity of the corresponding proteins at centrioles from three independent experiments. Representative images with enlargements of the centrosomal area are shown on the right. (C) Co-CM, *n* = 132; Wnt3a-CM, *n* = 115; (D) Co-CM, *n* = 150; Wnt3a-CM, *n* = 140; (E) Co-CM, *n* = 103; Wnt3a-CM, *n* = 112; (F) Co-CM, *n* = 122; Wnt3a-CM, *n* = 133. Scale bar: 10 µm. **(G, H)** RPE1 cells were treated with DMSO and CHIR99021 and serum starved for 16 h as depicted in Fig 1C. Cells were stained for ODF2 (G) and CEP83 (H). γ-tubulin served as a centriolar marker. DNA was stained with DAPI. The graph shows the fluorescence intensity of the corresponding proteins at centrioles from three independent experiments. Representative images with enlargements of the centrosomal area are shown on the right. (G) Control, *n* = 144; CHIR99021, *n* = 131; (H) Control, *n* = 136; CHIR99021, *n* = 127. Scale bar: 10 µm. *P* values are based on Student *t* test. The data underlying the graphs and blots in this figure can be found in the S2 Data and S1 Raw Images files.
(S4_Fig.TIF)

**S5 Fig. Inactivation of mTORC1 signaling rescues ciliogenesis upon Wnt signaling hyperactivation. (A, B)** RPE1 cells were treated with DMSO (control) or CHIR99021 for 16 h in the presence or absence of serum as depicted. Cells were stained for CEP164 (A) and CHIBBY (B). γ-tubulin served as a centriolar marker. The graph shows the fluorescence intensity of the corresponding proteins at centrosomes from three independent experiments. (A) Control, +serum, *n* = 118; Control, −serum, *n* = 101; CHIR99021, +serum, *n* = 187; CHIR99021, −serum, *n* = 125; (B) Control, +serum, *n* = 228; Control, −serum, *n* = 146; CHIR99021, +serum, *n* = 156; CHIR99021, −serum, *n* = 126. *P* values are based on Student *t* test. **(C)** Electron micrographs showing cross-sections of centrioles in RPE1 cells treated for 16 h with DMSO (control) or with the GSK3 inhibitors BIO and CHIR99021, as depicted. Two representative images per condition are shown. DA, distal appendages (blue asterisk), SDA, subdistal appendages (red asterisk). Scale bar: 200 nm. **(D, E)** U-ExM analysis of RPE1 cells treated with DMSO (control) or CHIR99021 for 16 h and stained for CEP164 (D) and CHIBBY (E). Acetylated tubulin antibodies were used to label centrioles. Representative images show enlarged top and side views of the centrosomal area. Scale bar: 250 nm. The data underlying the graphs in this figure can be found in the S2 Data file.
(S5_Fig.TIF)

**S6 Fig. Control of CP110 depletion.** Quantification of CP110 centrosomal levels of the experiment shown in Fig 4D and 4E for RPE1 cells treated with control (siLUC) or CP110-siRNA (siCP110). Antibodies against CP110, ODF2 and acetylated tubulin were used. The graph shows the fluorescence intensity of CP110 at centrioles from three independent experiments. siLUC, *n* = 101; siCP110, *n* = 109. P values are based on Student *t* test. The data underlying the graphs in this figure can be found in the S2 Data file.
(S6_Fig.TIF)

**S1 Table. List of antibodies used in this study.**
(S1_Table.XLSX)

**S2 Table. List of siRNAs used in this study.**
(S2_Table.XLSX)

**S1 Data. Raw data corresponding to the graphs in Figs 1–5.** The Excel table shows the datasets organized in separate tabs for each figure, with individual figure panels indicated within each tab.
(S1_Data.XLSX)

**S2 Data. Raw data corresponding to the graphs in S1-S6 Figs.** The Excel table shows the datasets organized in separate tabs for each figure, with individual figure panels indicated within each tab.
(S2_Data.XLSX)

**S1 Raw Images. Uncropped western blot membranes.** The PDF file shows the corresponding uncropped membranes to each figure panel, as indicated.
(S1_Raw_Images.PDF)

AcknowlegmentsWe thank Jon Lane (University of Bristol, UK) for providing the RPE1 mCherry-GFP-LC3 cell line and Oliver Gruss (University of Bonn, Germany) for generously sharing the PCM1 antibodies. We acknowledge Monica Langlotz from the ZMBH FACS Facility, Ulrike Engel from the Nikon Imaging Facility, and Felix Bestvater and Manuela Brom from the DKFZ Imaging Facility for their invaluable technical support and resources. We thank Elmar Schiebel for critical reading the manuscript.

## Author contributions

**Conceptualization:** Sergio P. Acebrón, Gislene Pereira.

**Data curation:** Cheng Yuan, Annett Neuner, Johanna Streubel.

**Formal analysis:** Cheng Yuan, Annett Neuner, Johanna Streubel, Gislene Pereira.

**Funding acquisition:** Matias Simons, Gislene Pereira.

**Investigation:** Cheng Yuan, Annett Neuner, Johanna Streubel.

**Methodology:** Cheng Yuan.

**Resources:** Ayushi Bhanushali, Sergio P. Acebrón.

**Supervision:** Gislene Pereira.

**Validation:** Cheng Yuan, Annett Neuner, Johanna Streubel.

**Visualization:** Cheng Yuan, Annett Neuner, Johanna Streubel.

**Writing – original draft:** Cheng Yuan, Gislene Pereira.

**Writing – review & editing:** Cheng Yuan, Annett Neuner, Johanna Streubel, Ayushi Bhanushali, Matias Simons, Sergio P. Acebrón, Gislene Pereira.

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
