## [Editor Report · Decision Letter 0]

22 Jan 2025

Dear Dr Pereira,

Thank you for submitting your manuscript entitled "Wnt/LRP6 signaling imbalance impairs ciliogenesis in human retina epithelial cells" for consideration as a Research Article by PLOS Biology.

Your manuscript has now been evaluated by the PLOS Biology editorial staff as well as by an academic editor with relevant expertise and I am writing to let you know that we would like to send your submission out for external peer review.

Once your full submission is complete, your paper will undergo a series of checks in preparation for peer review. After your manuscript has passed the checks it will be sent out for review. To provide the metadata for your submission, please Login to Editorial Manager (https://www.editorialmanager.com/pbiology) within two working days, i.e. by Jan 24 2025 11:59PM.

Kind regards,

Ines

--

Ines Alvarez-Garcia, PhD

Senior Editor

PLOS Biology

---

## [Decision Letter · Decision Letter 1]

11 Apr 2025

Dear Dr Pereira,

Thank you for your patience while your manuscript entitled "Wnt/LRP6 signaling imbalance impairs ciliogenesis in human retina epithelial cells" was peer-reviewed at PLOS Biology. Please also accept my apologies for the delay in sending you our decision. The manuscript has now been evaluated by the PLOS Biology editors, an Academic Editor with relevant expertise, and by two independent reviewers.

The reviews are attached below. As you will see, the reviewers find the conclusions interesting and worth pursuing, but they also raise several issues that would need to be addressed before we can consider the manuscript for publication. Reviewer 1 finds the title a bit misleading and thinks it should be improved, and also requests the addition of quantifications to some experiments along with an improved model that fits better the conclusions. Reviewer 2 suggests an experiment to analyse centriole ultrastructure under GSK3 inhibition using EM, and to measure CEP164 and Chibby levels on a time course prior to ciliogenesis with or without CHIR99021 treatment to check if GSK3 inhibition triggers active degradation of distal appendage components.

In light of the reviews, we would like to invite you to revise the work to thoroughly address the reviewers' reports. Given the extent of revision needed, we cannot make a decision about publication until we have seen the revised manuscript and your response to the reviewers' comments. Your revised manuscript is likely to be sent for further evaluation by all or a subset of the reviewers.

**IMPORTANT - SUBMITTING YOUR REVISION**

3. Resubmission Checklist

a) *PLOS Data Policy*

b) *Published Peer Review*

Sincerely,

Ines

--

Ines Alvarez-Garcia, PhD

Senior Editor

PLOS Biology

Reviewers' comments

Rev. 1:

The authors of this manuscript explore a highly important topic. While it has received significant attention in recent years, many previous studies have been disjointed and not carefully conducted. They show many different lines of evidence that high basal levels of Wnt signaling negatively affect ciliogenesis, and also begin to explore the mechanisms behind this. In particular they could show that early steps of ciliogenesis are blocked under high Wnt activity via satellite-bound OFD1 not being efficiently removed. In the final section of the results the authors show evidence to suggest that increased mTOR activity contributes to cilia loss under high Wnt baseline activation.

Compared to most Cilia WNT studies, experimental design has been carefully considered and shows that timing and cell line critically important. They also show that the observed effect is also seen in other cell lines.

The title is misleading. I would be careful with the wording that this work has conclusions for human RPE, since the RPE1 cells (we assume they are refereeing to the hTERT RPE1 cell line) are far removed from retinal pigment epithelial cells.

In general, better quantification of Immunos, WB and cilia lenth.

Since there are a variety of different cells lines being used. It would be very helpful to state clearly on the figures which cell line is being used when.

Specific Comments:

Since most of the non-ciliated cells still show a small extension of ARL13, I think one needs to be careful to call these non-ciliated. Its imperative that all cilia are also measured in terms of length.

Figure 1 A and B: No Results to show downregulation of these genes. Please provide evidence tos how levels of down regulation ideally on gene and protein level.

Figure 1 E: good example of figure that really needs to include quantification.

In Figure 1 and 2 Length of cilia not just presence of cilia also important. Why not include this?

Figure 3a: Why is difference between control and Wnt-3a getting smaller over time? Is Wnt induction by wnt-3a is decreasing over time. Maybe the authors can check Beta Catenin expression for other time points as they did for 16h.

Sup. fig 1H: Nice to show Wnt surrogate affect, but from these comparisons it looks as if Wnt surrogate induces proliferation (due to more cells).

Figure 2B: Quantification of this figure would be really important. Actin seems thicker for siTCF7 wnt3a-cm. So its difficult to say that TCF7 has no effect on hyperphosphorylated forms of DVL2.

Figure 3 G-F-H: For completions sake, would it be possible to include Wnt3a for Cep164 and Chibby and CHIR for ODF2.

Figure 4 A: Typo MotheR centriole

Fig. 2 E Supplement: What is happening with Cep83? The P value looks significant 0.0087. Is the number wrong? In the text they state that this is not significant?

Figure 4D: Can you quantify downregulation of CP110.

Final Model is not a good summary of the findings and difficult to interpret: Wnt induction decreases cilia number and when Wnt is active, GSK3 is inhibited. However, the authors also show that GSK3 inhibition increases ciliation, therefore based on this model (as depicted) Wnt should also increase ciliation.

Rev. 2:

This manuscript by Yuan and colleagues investigates the impact of Wnt/LRP6 signaling on ciliogenesis in RPE1 cells. The authors primarily employ Wnt3a-conditioned media, GSK3 and mTOR inhibition, depletion of Wnt pathway components, and centriolar/cilia components, to explore the underlying mechanisms. Overall, the results are compelling, and the observed defects in ciliogenesis are convincingly demonstrated. The authors propose that Wnt activation impairs the removal of the cilia inhibitory protein OFD1, and that mTOR inhibition can rescue cilia formation under Wnt-activated conditions.

However, I find the proposed mechanism somewhat unclear, and several points require clarification prior to publication:

1: As discussed by the authors, GSK3 inhibition (but not Wnt3a treatment) reduces the levels of distal appendage proteins CEP164 and Chibby at mother centrioles, implying a structural defect. It would be valuable to examine centriole ultrastructure under GSK3 inhibition using electron microscopy. Although the loss of Chibby is less pronounced in Wnt3a-treated cells, there still appears to be a reduction. It would strengthen the study to use super-resolution microscopy to assess whether the spatial organization of CEP164 and Chibby is preserved or not under Wnt3a treatment.

2: Can the authors measure CEP164 and Chibby levels over a time course prior to ciliogenesis, with and without CHIR99021 treatment? Specifically, do these proteins increase during serum starvation, or are their levels stable and then reduced upon GSK3 inhibition? This would provide insight into whether GSK3 inhibition triggers active degradation of distal appendage components.

3: For all experiments, it is unclear whether the observed impairment in ciliogenesis represents a complete block or a delay. A brief clarification in the text would help readers better interpret the results.

4: Figure 1, there are noticeable differences in the percentage of ciliated cells across control experiments. Could the authors explain this variability, as it may influence statistical comparisons ?

5: A brief introduction to TCF7 and its relevance in this context would be helpful for readers unfamiliar with its role in Wnt signaling.

6: For panels 3B-D, please use consistent color coding for protein staining to facilitate comparison.

7: The authors state, "Collectively, the data indicate that early steps of ciliogenesis are blocked under high Wnt activity." Is the process entirely blocked, or is it delayed? Please clarify.

8: I may have missed this data, but would it be possible to quantify ODF2 intensity under CHIR99021 treatment?

9: Figure 4 : In the CP110 siRNA experiments, did the authors observe microtubule elongation phenotypes, even in the absence of full ciliogenesis? This phenotype is often independent of cilium formation.

10: In the OFD1 knockdown experiment, the percentage of ciliated cells appears low in both control and treated conditions. Could this reflect an early time point, consistently with the fact that Wnt signaling delay rather than completely block ciliogenesis (and what is the result at later time point)? If delay is the major phenotype, the authors might consider adjusting the title to reflect this nuance—for example, by using the term 'delayed ciliogenesis'

---

## [Editor Report · Decision Letter 2]

11 Aug 2025

Dear Dr Pereira,

Thank you for your patience while we considered your revised manuscript entitled "Hyperactivation of Wnt/LRP6 signaling delays ciliogenesis in hTERT-RPE1 cells" for publication as a Research Article at PLOS Biology. This revised version of your manuscript has been evaluated by the PLOS Biology editors and the Academic Editor.

Based on our Academic Editor's assessment of your revision, we are likely to accept this manuscript for publication, provided you satisfactorily address the data and other policy-related requests stated below my signature.

In addition, we would like you to consider a suggestion to improve the title:

"The interplay between Wnt and mTOR signaling modulates ciliogenesis in human retinal epithelial cells"

We expect to receive your revised manuscript within two weeks.

*Published Peer Review History*

*Press*

Sincerely,

Ines

--

Ines Alvarez-Garcia, PhD

Senior Editor

PLOS Biology

DATA POLICY:

Thank you for providing the data underlying the graphs shown in the figures. I have checked them all and I have the following queries:

1. The data underlying Fig. S1C graph is labelled as Fig. S1B in the S2 Data file, please correct this.

2. The data underlying Fig. S2C graph doesn’t seem to correspond to the data shown in the graph – there are 7 sets of values, but only 6 sets in the figures. Please check the data and indicate the hours.

CODE POLICY

Many thanks for providing the original raw gels shown in the figures. Please address the following points:

1. The raw gel shown for Fig. 1D is labelled as Fig. S2B and seems cropped. Please add the full raw gel and re-label it in the file.

2. The raw gel shown for Fig. S1A is labelled as Fig. 1A, please re-label it.

3. The raw gel shown for Fig. S1B is labelled as Fig. 1B, please re-label it.

4. The raw gel shown for Fig. S2B is labelled as Fig. S2C, please re-label it.

5. The raw gels for Fig. 4G, Fig. 5C and Fig. S4A are missing. Please provide them in the file.

Please carefully read our guidelines for how to prepare and upload this data: https://journals.plos.org/plosbiology/s/figures#loc-blot-and-gel-reporting-requirements

---

## [Editor Report · Decision Letter 3]

14 Aug 2025

Dear Dr Pereira,

Thank you for the submission of your revised Research Article "The interplay between Wnt and mTOR signaling modulates ciliogenesis in human retinal epithelial cells" for publication in PLOS Biology. Please note that I am sending this letter on behalf of my colleague Ines Alvarez-Garcia since she is currently away from the office this week. On behalf of my colleagues and the Academic Editor, Dagmar Wachten, I am pleased to say that we can in principle accept your manuscript for publication, provided you address any remaining formatting and reporting issues. These will be detailed in an email you should receive within 2-3 business days from our colleagues in the journal operations team; no action is required from you until then. Please note that we will not be able to formally accept your manuscript and schedule it for publication until you have completed any requested changes.

PRESS

Best wishes, 

Richard

Richard Hodge, PhD

rhodge@plos.org

On behalf of:

Ines Alvarez-Garcia, PhD

PLOS
